# In-vivo studies on Transitmycin, a potent *Mycobacterium tuberculosis* inhibitor

Rajesh Mondal[1], Azger Dusthackeer V. N.[2]*, Palaniyandi Kannan[2], Amit Kumar Singh[3], Kannan Thiruvengadam[2], Radhakrishnan Manikkam[4], Shainaba A. S.[2], Mahizhaveni Balasubramanian[2], Padmasini Elango[2], Sam Ebenezer Rajadas[2], Dinesh Bharadwaj[5], Gandarvakottai Senthilkumar Arumugam[6], Suresh Ganesan[6], Hemanth Kumar A. K.[2], Manjula Singh[7], Shripad Patil[3], Jaleel U. C. A.[8], Mukesh Doble[9], Balagurunathan R.[10], Srikanth Prasad Tripathy[11], Vanaja Kumar[11]

1 ICMR - Bhopal Memorial Hospital & Research Center, Bhopal, Madhya Pradesh, 2 ICMR-National Institute for Research in Tuberculosis, Chennai, India, 3 ICMR-National JALMA Institute for Leprosy & Other Mycobacterial Diseases, Agra, Uttar Pradesh, 4 Sathyabama Institute of Science and Technology, Chennai, India, 5 ICMR- National Institute of Nutrition, Hyderabad, Telangana, 6 Indian Institute of Technology Madras, Chennai, India, 7 ICMR-ITRC, NewDelhi, India, 8 OSPF NIAS Drug Discovery Lab, National Institute of Advanced Studies, Indian Institute of Science Campus, Bangalore, India, 9 Saveetha Dental College and Hospitals, Chennai, India, 10 Periyar University, Salem, India, 11 Ex-ICMR-NIRT, Chennai Scientists, Chennai, India

* azgerdusthackeer.vn@icmr.gov.in

**Data Availability Statement:** All relevant data are within the paper and its Supporting information files.

## Abstract

This study involves the *in-vitro* and *in-vivo* anti-TB potency and *in-vivo* safety of Transitmycin (TR) (PubChem CID:90659753)- identified to be a novel secondary metabolite derived from *Streptomyces* sp (R2). TR was tested *in-vitro* against drug resistant TB clinical isolates (n = 49). 94% of DR-TB strains (n = 49) were inhibited by TR at 10μg ml$^{-1}$. *In-vivo* safety and efficacy studies showed that 0.005mg kg$^{-1}$ of TR is toxic to mice, rats and guinea pigs, while 0.001mg kg$^{-1}$ is safe, infection load did not reduce. TR is a potent DNA intercalator and also targets RecA and methionine aminopeptidases of *Mycobacterium*. Analogue 47 of TR was designed using *in-silico* based molecule detoxification approaches and SAR analysis. The multiple targeting nature of the TR brightens the chances of the analogues of TR to be a potent TB therapeutic molecule even though the parental compound is toxic. Analog 47 of TR is proposed to have non-DNA intercalating property and lesser *in-vivo* toxicity with high functional potency. This study attempts to develop a novel anti-TB molecule from microbial sources. Though the parental compound is toxic, its analogs are designed to be safe through *in-silico* approaches. However, further laboratory validations on this claim need to be carried out before labelling it as a promising anti-TB molecule.

## Introduction

Tuberculosis (TB), an age-old bacterial disease that has established its presence on the face of this Earth for many myriad years successfully. Anti-tuberculosis therapy started flourishing with the discovery of Streptomycin (S) from *Streptomyces griseus* by Albert Schatz, Selman

**Funding:** This study was funded by Indian Council of Medical Research for providing grant-in-aid (F. No.5/8/5/8/TF/2017/ECD-I, to conduct this preclinical study on Transitmycin. The funders had no role in study design, data collection and analysis, decision to publish, or preparation of the manuscript.

**Competing interests:** Authors have declared that no competing interests exist.

Waksman, and Elizabeth Bugie in 1943 [1]. Para-amino salt of salicylic acid (PAS) discovered by Jorgen Lehmanin in the same year was also found efficient against TB. British Medical Research Council (BMRC) in the year 1950 recommended the prescription of these two drugs to TB patients. Later from 1952 onwards, other drugs including isoniazid (INH), pyrazinamide (PZA), ethambutol (EMB), and rifampicin (RIF) were added to the regimen that reduced the treatment period from 18 months to 6 months [2, 3]. Turbulence occurred in this peaceful stream of TB therapy with the emergence of drug-resistant *Mycobacterium tuberculosis* (Mtb) between 1990 and 2000 [4, 5] advocating the need for new anti-TB drugs. The Food and Drug Administration (FDA) in 2012 and World Health Organization (WHO) in 2014 have recommended bedaquiline and delamanid respectively to treat drug-resistant TB cases [6, 7]. Unceremoniously resistance to bedaquiline and delamanid was also observed within a year of administration [8]. Hence, still, there is an urge for newer anti-TB drugs that are safe and potent.

Kumar *et al.*, (Patent number US-2016200769-A1, PubChem CID:90659753) has reported a novel drug named transitmycin (TR) with anti-mycobacterial potential. TR is a secondary metabolite produced by *Streptomyces* sp. R2 (MTCC5597; DSM26035). Interestingly, TR was found efficient against the clinical and standard laboratory strains of Mtb, including drug-resistant forms. TR was found to be virucidal against recombinant Human Immunodeficiency Virus (HIV) while the same is non-cytotoxic on mouse embryo fibroblast cell lines. These characteristics of TR projected it to be a promising anti-TB Lead and hence carried forward for further investigation on its *in-vivo* efficacy and cytotoxicity and drug target prediction (*in-vivo* and *in-vitro*). This article describes our experiences gained in the pre-clinical phase while validating TR for its anti-MDR-TB potential.

## Materials and methods

### Ethics statement

The study was approved by the ICMR-NIRT IEC (001/NIRT-IEC/2019), ICMR-NIN animal ethics (NCLAS/IAEC/01/2017/P3F) and by the ICMR- JALMA IAEC (NJIL&OMD/3-IAEC/2019-01).

### Animals and bacterial strains

*Streptomyces* sp. R2 (MTCC5597; DSM26035) (hereafter mentioned as R2) was maintained as slant stock in yeast extract & malt extract (YEME-Himedia, India) agar comprised of 50% seawater, glycerol stock in 30% glycerol as well as in lyophilized form. Cultures were revived on fresh YEME agar plates after incubation for 7–10 days at 28˚C. The crude ethyl acetate extracts from R2 grown agar media were purified by adopting the preparative HPLC to yield TR with more than 98% purity. Mycobacterial clinical and laboratory standard strain H37Rv were grown on Lowenstein Jenson (LJ) egg-based media. While testing the in-vitro anti-TB Middlebrook 7H9 liquid media supplemented with oleic acid, albumin, dextrose, and catalase (OADC) (BD Biosciences) were used. All the Mtb cultures were incubated at 37˚C unless stated otherwise.

Balb/c / Swiss Albino Mice weighing 20–25g, Duncan Hartley Guinea pigs weighing 350–400g and Sprague Dawley rats weighing 180–200g were used. In all the experiments, an equal number of male and female animals were used. The animals were maintained at 24 ± 2˚C, 50 to 60% relative humidity, with a 12 hours light-dark cycle. All the animals were acclimatized to laboratory conditions, at least 7 days before conducting tests. Entire *in-vivo* studies were carried out in the BSL3 animal house of the National JALMA Institute for Leprosy and Other Mycobacterial Diseases, Agra, India. Each group of animals was kept in separate isolators to

prevent cross-infection between the animals. This study was approved by the Institutional Ethics Committee (IEC) of the National Institute for Research in Tuberculosis (001/NIRT-IEC/2019 dt.02.01.2019) and Institutional Animal Ethics committee of National JALMA Institute for Leprosy and Other Mycobacterial Diseases (NJIL&OMD/3-IAEC/2019-01 dt. 26.07.2019).

### Extraction and purification of TR

The crude ethyl acetate extract (R2) containing TR was subjected to column chromatography using neutral alumina with Chloroform (Sigma) and Methanol (99%:1%) (Sigma) as eluent. Fractions were collected and concentrated under vacuum to attain 95% purity. This was repeated once again for further purification. Screening of the eluted fractions was done through thin-layer chromatography with a pre-coated alumina-silica sheet. The purified product (TR) was analyzed using [1]H NMR, ESIMS Mass spectra and HPLC.

### Synthesis of TR analogs

TR is structurally identical to the known anti-cancer drug; actinomycin, except in the valine motif and additional oxygen (Fig 1). Synthesis of TR analogs was carried out by following the methods as described elsewhere for the synthesis of actinomycin drug analogs [9–12]. Analogs

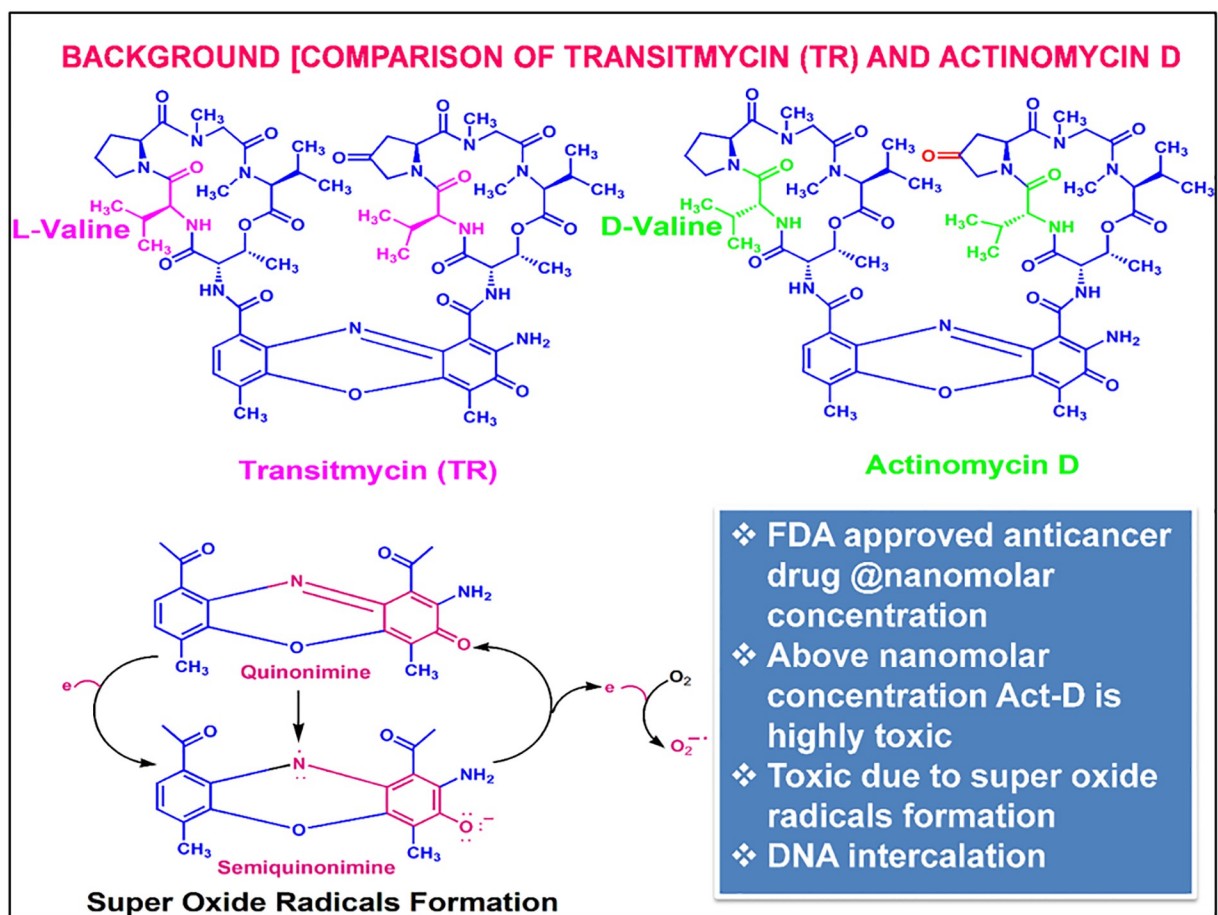

**Fig 1. Structural comparison between TR and actinomycin.**

of the parent compound were synthesized by deactivation of $NH_2$ group by N-substitution via chemical modification and deactivation of the keto group by chemical reduction methods.

Analogs of TR namely, TR-NC6, Tr-SUND, TRR, TR-RB, FN-Me, and C-tertbutyl were synthesized. TR-NC6 was synthesized by N-Alkylation of TR using 4-Hydroxy (hexyl)-Coumarin. Tr-SUND was derived by N-Glycosylation of TR using Acetobromo-α-D-Glucose. TRR was prepared by sodium borohydride ($NaBH_4$) mediated reduction of TR's keto group. Benzoylation of the TR OH group using 4-Methoxy-Benzoyl Chloride resulted in the TR-RB. N-Alkylation of TR using Iodomethane was done to synthesize FN-Me and Bromination-alkylation of TR using 3,5-Ditert-buty-4-hydroxybenzoic acid methyl ester was used to prepare C-terbutyl-TR (S14 Fig in S1 File).

## Screening of *in-vitro* anti-Mtb activity of TR and its analogs

The Minimum Inhibitory Concentration (MIC) and Minimum Bactericidal Concentration (MBC) of TR against drug resistant clinical isolates and laboratory strains of Mtb were tested *in-vitro*.

The MIC of TR against individual Mtb clinical isolates and laboratory strain (H37Rv) was evaluated in comparison to the other known drugs, isoniazid, rifampicin, kanamycin, moxifloxacin, bedaquiline and delamanid. MIC determination was carried out using the Broth-microdilution method. The 1.0 McFarland matched suspension of Mtb culture that was grown on LJ slope was further diluted in a 1:19 ratio using 7H9 broth and used to estimate the MIC using the broth micro-dilution method as described elsewhere [13]. In brief, the TR and other standard drugs were taken in 96 well Microtiter plates in the specified range of concentrations. Each well was loaded with 100μl of diluted Mtb culture suspension to arrive at a final volume of 200μl. Separate culture controls and solvent controls were included to ensure the culture viability. Post 14 days of incubation at 37˚C, the plates were observed under an inverted phase-contrast microscope for the Mtb characteristic serpentine cord formation. The least concentration of the drug that resulted in the visible absence of growth was accounted for as its MIC. All the tests were carried out in duplicates and the absence of microbial contamination was ensured by spotting the aliquots from the plate onto Brain Heart Infusion (BHI) agar plates.

To determine the MBC of TR, cell suspension of Mtb clinical isolates (MDR and XDR) and laboratory strain (H37Rv) equivalent to 1.0 McFarland standard was taken and further diluted in 1:19 ratio using 7H9 media. The cell suspension was taken in 96 well Microtiter plate and treated with different drug concentrations. After 3 days of incubation, 5μl cell suspension from each well was spotted onto 7H11 agar plate in triplicates and incubated until the growth appears. The minimum concentration with no growth in all of the triplicates was considered as MBC.

## Screening of *in-vitro* anti-Mtb activity analogs of TR using Luciferase Reporter Phage (LRP) assay

The anti-Mtb activity of TR, its MIC and its analogs such as TR, Compound 1 (R); Compound 2 (RB); compound 3 (TrSUND), and compound 4 (TrNC6) were performed by LRP assay along with the laboratory strain H37Rv by following the NIRT in-house developed protocol as described elsewhere [14]. LRP assay is a method which uses a genetically engineered mycobacteriophage for detecting the presence of live *Mycobacterium tuberculosis* present in the cultures and is used as a screening procedure for detecting the presence of live cells of tubercle bacilli in the presence of Transitmycin.

## Pharmacokinetic study on TR

Pharmacokinetics of plasma TR concentration was estimated using the methods as described elsewhere [15]. In brief, the pooled human plasma obtained from blood bank, Chennai as followed previously not containing Tr was used to prepare Tr concentrations of 0.75, 3.75, 10.5 and 15µg ml$^{-1}$. These samples were spiked with known concentrations of Tr. Based on this standardized protocol, we subjected the same experiment with guinea pig models to estimate the Pk (Pharmacokinetics) values on TR.

## *In-vivo* estimation of Maximum Tolerated Dose (MTD)

Determination of Lethal Dose (LD50) and MTD was carried out in Balb/c Mice, Sprague Dawley rat and Dunkin Hartley guinea pigs. All the animals used in this study were maintained in the laboratory conditions as per the CPCSEA guidelines.

Balb/c / Swiss Albino Mice (n = 32) were randomly divided into groups with an equal gender ratio as mentioned in S1 Table in S1 File. The animals were maintained at 24 ± 2˚C, 50–60% relative humidity, with a 12h light-dark cycle. Either via intravenous route (IV) or intraperitoneal (IP) route each group was administered with a single dose of TR. The parental route of administration was selected since the compound is a peptide in nature (Route of administration is given in S1 Table in S1 File).

The test compound (TR) is being investigated for the first time *in-vivo* system. Based on the previous reports on the *in-vitro* activity that stated 6.25µg ml$^{-1}$ of TR as its MIC (minimum inhibitory concentration) and by comparing it with rifampicin or streptomycin, the dosage has been tittered and administered. Doses are 0.15, 0.3 and 0.45mg kg$^{-1}$ through IV for mice.

The Guinea pigs with normal health reports were acclimatized for 1 day in the experimental room. This was followed by administration of the test compound to a respective group of animals through IP as 3 doses on alternate days. The maximum volume of administration is 0.5ml per guinea pig (S2 Table in S1 File).

The Sprague Dawley rats and WNIN rats with normal health reports were conditioned for 3 days in the experimental room. This was followed by test compound administration Intravenously as a single dose or twice by Intramuscular route to the respective group of animals. The maximum volume of administration was 0.5ml per rat (S3 Table in S1 File).

The animals were observed for lethality 3$^{rd}$, 6$^{th}$, 12$^{th}$& 24$^{th}$ hour after test compound exposure, general behavior and daily activity. Routine physical, physiological examination, and neurological activity were done bi-weekly till the end of the experiment. In the event of severe toxicity, animals showing adverse effects were euthanized. These included effects on body weight (body weight loss >10%), reduced food consumption and subdued behaviour [16]. In the event of the death of any animal during the observation period, an autopsy was conducted and all vital organs (brain, liver, kidney, heart, lung, reproductive organs & intestine) were collected for studying histopathological changes.

## Determination of acute toxicity

An acute toxicity study was conducted in mice and Guinea pigs with a single exposure at 10–50 times of intended therapeutic dose to determine the activity and mortality of the test compound based on the LD50 levels that were determined from the MTD study.

Thirty-six Balb/c / Swiss Albino Mice (20–25g) and thirty-six Dunkin Hartley Guinea pigs (350–400g) were divided into three groups (n = 12 in each group with an equal gender ratio). Each group was exposed to a single dose of TR through IV (S1 and S2 Tables in S1 File).

## Safety study in guinea pig

The experiments involving use of guinea pigs were approved by Institutional Animal Ethics Committee of ICMR-National JALMA Institute for Leprosy & Other Mycobacterial Diseases, Agra, India. Three male guinea pigs were included in the study and were treated with the TR at further lower doses i.e. at $0.001 mgkg^{-1}$ at alternate days for 30 days (Total 15 dose) (S1 Fig in S1 File). The drug was prepared as described previously and injectable volume was kept constant at 500µl. The 400µl blood was collected for biochemical analysis by retro-orbital puncture under the influence of isoflorane anesthesia prior to start of treatment and after 10[th] and 15[th] dose of TR. One guinea pig was picked randomly and sacrificed at 3[rd] day, 15[th] day and 45[th] day after last injection of TR. The animals were euthanized by overdose of xylazine and ketamine and lung, spleen, kidney, heart and liver were removed aseptically for gross and histopathological examination in 10% buffered formalin.

## *In vivo* efficacy studies on TR in guinea pig model of tuberculosis

The male outbred Hartley guinea pigs with weight ranging between 290-330g were purchased from Animal house facility of the Lala Lajpat Rai University of Veterinary and Animal Sciences (*LUVAS*), Hisar, and were held under barrier conditions in a Biosafety Level-3 animal laboratory during the entire experiment duration. All guinea pigs were treated in accordance with Committee for the Purpose of Control and Supervision of Experiments on Animals (CPCSEA) guidelines and approved protocols. After allowing the animal to acclimatize, 42 animals were infected with a low dose Mtb H37Rv via aerosol route using Glas-Col chamber aerosol generation device (Glas-Col, USA). After infection, each guinea pig was returned to its home cage, and monitored daily for changes in weight or body condition. All animals were routinely cared for as per the guidelines prescribed by the CPCSEA for Laboratory Animal Care which annually inspects all facilities to certify compliance with highest possible levels of housing conditions, feeding regimens, and environmental enrichment.

Bacterial loads in the organs of guinea pigs (*n = 3*) were determined a day after infection to determine the number of implanted bacilli in guinea pigs' lung. The remaining animals were randomly assigned to seven groups (N = 6group$^{-1}$) (S4 Table & S2 Fig in S1 File). On day 15[th] of the infection, six guinea pigs were sacrificed to determine the bacterial load prior to the start of treatment (Early Control). The remaining animals were randomly assigned to four groups: an untreated control group (Late Control), a group receiving the rifampicin (RIF), and isoniazid (INH) treatment (Positive Control) or erythromycin (negative control). Animals in the antibiotic control group were treated with rifampicin (RIF; 50 mg kg$^{-1}$ of mean body weight), and isoniazid (INH; 30 mg kg$^{-1}$ of mean body weight) by oral gavage [17]. A split feeding protocol was used for INH & Rif group, with INH being given in the morning and RIF being given an hour later to minimize drug interactions. The erythromycin was administered at the rate of 50 mg kg$^{-1}$ daily by oral gavage. Another group of guinea pigs received TR treatment at different dose rate (0.01, 0.02 or 0.04 mg/kg respectively) every alternate day by IP route.

The final DMSO concentration in TR injectable solution was 0.001%. As it was an exploratory study and considering the toxicity and non-availability of pharmacokinetics data, the dosing schedule and route of administration were kept similar to toxicity studies i.e. intraperitoneal injection at 48hrs interval. Guinea pig health was checked for smoothness of the fur, movement and activity, and reaction to outside stimuli. Guinea pigs were weighed every alternate day before dosing for dose adjustment. Weight loss of 15% to 20% has been the most commonly cited endpoint for tuberculosis studies. Humane endpoints were predetermined in the study protocol and guinea pigs were euthanized when their weight loss was >20% [18]. Bacterial loads in the organs of guinea pigs were determined after 3[rd], 5[th] and 7[th] dose of TR as

showed in S2 Fig in S1 File. Treatments was stopped 2 days before necropsy in randomly selected animals (2 in number) from each group and was euthanized by overdose of xylazine and ketamine and were sacrificed under aseptic conditions. The lungs, liver and spleen were aseptically removed, examined and photographed for gross pathology. Lungs and spleen were homogenized and plated on Middlebrook 7H11 plates for CFU enumeration. The bacterial colonies counted after 3 to 4 weeks of incubation at 37°C. Further, the left caudal lung from animals was fixed in 10% buffered formaldehyde for histopathology studies.

### Determination of host biochemical response to TR

The blood from TR treated guinea pigs were collected via retro-orbital puncture on day 7, 12, and 15 post-TR administration. Samples for clinical chemistry parameters were assayed on the Hitachi 902 auto-analyzer (Japan). The parameters examined were: Alanine aminotransferase (ALT), aspartate aminotransferase (AST), Serum creatinine, blood urea, and a serum cholesterol level that are commonly used as blood markers to monitor liver and kidney functions. After treatment completion, the blood was collected from surviving guinea pigs every 10[th] day after the last dose of TR till the day of sacrifice.

### TR target identification and elucidation of the mechanism of action

Identification of TR target in the mycobacterial system was analyzed *in-silico* using machine learning and cheminformatics-based analysis. Schrodinger glide and Canvas 2, Autodock vina, Ochem server and algorithms, and Cheminformatics bioinformatics & AI tools are used for this purposes.

### EtBr displacement assay for *in-vitro* detection of Mtb DNA intercalation

To determine the DNA intercalating potential of the TR and its analogs ethidium bromide displacement assay was performed by following the method as described previously with few modifications [19]. In brief, varying concentrations of the same ranging from 0.1 to 5μg ml$^{-1}$ treated a defined concentration of Mtb chromosomal DNA (200ngμl$^{-1}$). This TR and its analogs treated DNA was exposed to 0.126mM ethidium bromide (EtBr) (Sigma). This was done to estimate the competitive intercalation of EtBr with TR and its analogs into the groves of Mtb DNA. Fluorescence intensity was observed through excitation at 546nm (Multimode-mode plate reader, Spinco Biotech) and emission at 595nm. The percentage of reduction in relative fluorescence (RFU) was calculated using the formula given below. The fluorescence intensity is inversely proportional to the interaction of the test compounds with the Mtb DNA.

$$Percentage\ of\ Reduction\ in\ RFU = \left( \frac{RFU\ of\ Control - RFU\ of\ test}{RFU\ of\ control} \right) x\ 100$$

## Results

### Extraction and purification of TR

The biologically potent molecule (TR) was isolated as well as characterized from the crude samples by using Thin-layer Chromatography (TLC), column chromatography, $^1$H NMR, HR ESIMS, and HPLC sequentially. The purity of the TR was ascertained to be 98.64%. Detailed chromatograms are given in the S1 File.

### Determination of the critical concentration of TR using LRP assay

Minimal inhibitory concentration of TR ranging from 5, 10 and 50 μg ml$^{-1}$ was determined using luciferase reporter phage assay (LRP) (S5 Table in S1 File). The critical concentration (CC) of TR was deduced to be 10μg ml$^{-1}$. The CC is defined as the lowest concentration of an anti-TB drug capable of inhibiting the growth of at least 95% of wild-type *M. tuberculosis* in vitro in solid media (Lowenstein-Jensen; LJ, slopes) or liquid medium (mycobacteria growth indicator tubes; MGIT) [20].

### *In-vitro* anti-TB efficacy of TR and its analogs

MDR and XDR TB clinical isolates (n = 50) obtained from culture repository, National Institute for Research in Tuberculosis, Chennai as blind controls. The MIC was performed for the concentration of transitmycin between 0.312μg ml$^{-1}$ and 40μg ml$^{-1}$, along with isoniazid (INH), rifampicin (RIF), bedaquiline (BDQ), kanamycin (KAN), moxifloxacin (MXF) and delamanid (DEL).

This figure shows that the observed MIC of Transitmycin differs [d1] significantly (p<0.001) from the standard drug critical concentration in the same drug-resistant clinical isolates (Fig 2). The difference of Transitmycin MIC is lower than the standard critical concentration compared to the differences observed in other drugs of the same clinical isolates. The paired t-test was used to evaluate this statistical differences [d1].

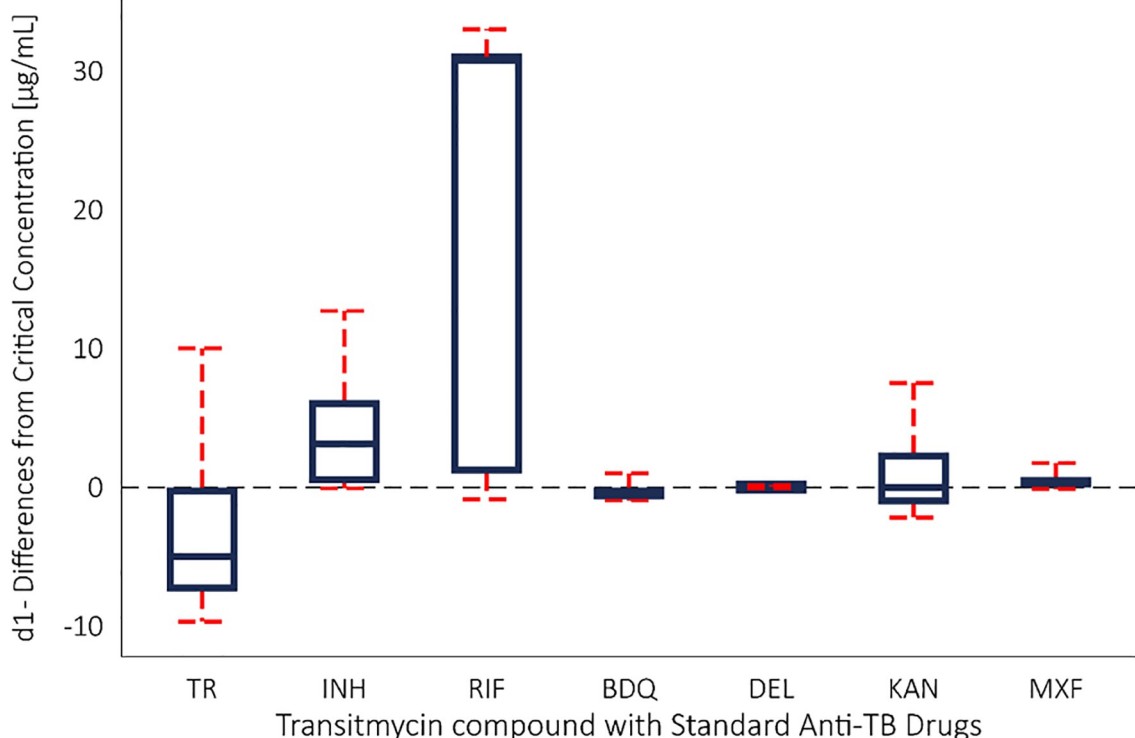

1. CC of the drugs- Transitmycin [TR-10μg/mL]; Isoniazid [INH-0.1μg/mL]; Rifampicin [RIF-1μg/mL]; Bedaquiline [BDQ-1μg/mL]; Delamanid [DEL-0.06μg/mL]; Kanamycin [KAN-2.5μg/mL]; Moxifloxacin [MXF-0.25μg/mL]
2. Transitmycin is signicantly different when compared to other drugs based on Critical Concentration (CC)

**Fig 2. The extent that the MICs (n = 50) differ from the observed critical concentration (CC) of TR in comparison to INH, RIF, BDQ, KAN, MXF, and DEL.**

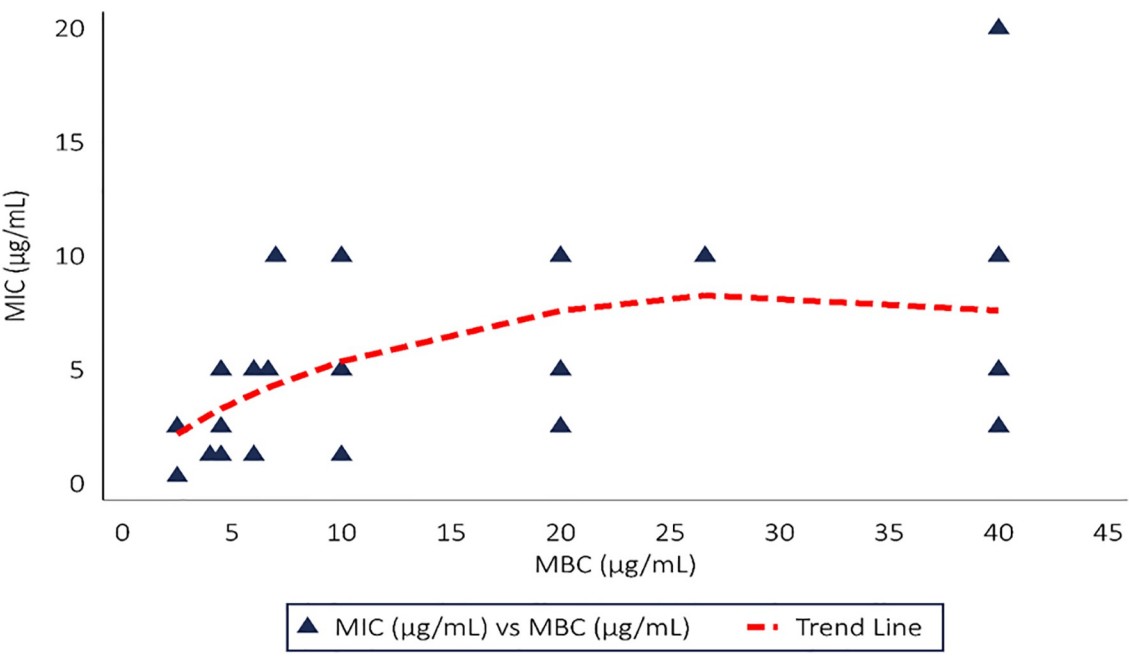

Correlation: 47.7%; p=0.002
The average difference between MIC (µg/mL) vs MBC (µg/mL) is 12.8 (8.7- 16.9); p<0.001

**Fig 3. MIC and MBCs of TR in 39 drug-resistant isolates of *M. tuberculosis*.**

The MIC and MBC for TR was determined and was observed to be significantly correlated (correlation = 47.7%, p = 0.002) where the correlation was assessed by Pearsons correlation. The observed values of the MBC were significantly higher than those of the MIC values with a difference of 12.8µg/mL (8.7µg/mL–16.9µg/mL), p0.001. The difference between them was tested with a one sample t-test assuming that they are similar and do not differ (Fig 3).

The analogs of TR; TR-SUND, TR-RB, TR-R, TR-NC6, F-N Me, and Ct-butyl was screened for anti-TB activity using Luciferase reporter Phage (LRP) assay. Among these seven analogs, only F-N Me showed maximum inhibition (77.30%) at 40µg ml$^{-1}$ concentration. However, TR at the same concentration inhibited 98.40% of Mtb H37Rv (S6 Table in S1 File).

## *In-vivo* safety analysis on TR

**MTD determination.** In the mice, mortality was observed till the concentration of 0.45mg kg$^{-1}$ dose on the 7th day and the 11th day of post-exposure via IV route. The mouse survived at the dosage of 0.15 and 0.3mg kg$^{-1}$ and was found to be partially active at the 30$^{th}$ day post-drug exposure. Elevated levels of enzymes that indicate the hepato-renal toxicity was observed with the potential of local necrosis at the site of injection. In the rats, all the administered doses of 0.1, 0.2 and 0.4mg kg$^{-1}$ are found to be toxic with mortality and hepato-renal damages. Meanwhile, IV administration of TR in the rats at 0.025, 0.05 & 0.1mg kg$^{-1}$ dosages showed no mortality, but local necrotic lesions at the site of injection. IM route of TR administration at 0.025 and 0.05mg kg$^{-1}$ dosage level caused zero mortality up to 16$^{th}$-day post-drug injection. However, 0.1mg kg$^{-1}$ dose resulted in mortality on the 10$^{th}$ day of TR administration. In the guinea pigs, 0.025mg kg$^{-1}$ dose via IP route was highly toxic. However, the dose of 0.001 mg kg$^{-1}$ in uninfected guinea pigs was noted to be safe and there was no loss in body weight and appetite over the study period (S3 Fig in S1 File).

**Gross pathology during treatment.** Visual hemorrhage could be seen in the lungs of guinea pigs treated with TR @ 0.01, 0.02 or 0.04 mg/kg body weight respectively. and The spleen size was also reduced. Further, fibrous-like growth could be visualized over the liver in the TR group and all the lobes adhered to each other. Hematoxylin and Eosin (H&E) stained sections of the lungs and the heart showed lymphocyte infiltrations. Histopathological examinations of the spleen, liver, heart, and kidney of the uninfected guinea pigs treated with TR @ 0.001 mg/kg showed no abnormality at all time points of the study, however, lymphocytic infiltration around bronchioles was seen at all time points (S4 Fig in S1 File).

**Effect of TR on host biochemical parameters.** Plasma and serum samples were collected and analyzed for liver and kidney function tests and cholesterol level analysis. The results show a significant increase in blood urea, creatinine following treatment with TR at 0.02 and 0.04mg kg$^{-1}$ respectively, indicative of impaired kidney function. Similarly, the increased level of AST and ALP were observed for TR at 0.04mg kg$^{-1}$ dose, indicative of liver damage (>700 U/L and >350U/L respectively) (S5 Fig in S1 File). Likewise, a significant increase in blood cholesterol (more than 3 fold increase) was also observed in TR-treated guinea pigs at all dose ranges. The variations in serum enzyme levels (SGOT, SGPT, Serum Creatinine, Blood and Urea) at 0.001mg kg$^{-1}$ dose were within the reference range of guinea pigs and there were no marked alternation suggestive of toxicity during the study period.

**Effect of lower dose of TR on in-vivo safety.** TR at the concentration of 0.01mg kg$^{-1}$ dosage showed higher toxic effects in the guinea pigs with no significant variation in the infection load when compared to the INH+RIF treated group. Even the separate group that was exposed to 0.005mg kg$^{-1}$ dose displayed similar pathology as it was in the previous groups. Hence an attempt was made to explore the possibilities of using TR at a lower concentration of 0.001mg kg$^{-1}$ (S1 Fig in S1 File).

There was no loss in body weight and appetite over the study period at 0.001 mg/kg in uninfected guinea pigs. The weight increased throughout the experiment duration (S3 Fig in S1 File). The variations in serum enzyme levels (SGOT, SGPT, Serum Creatinine, Blood and Urea) were within the reference range of guinea pigs and there were no marked alternation suggestive of toxicity during the study period (S5 Fig in S1 File). Histopathological examinations showed no abnormality at all time points of the study in the spleen, liver, heart, and kidney of the uninfected group however, lymphocytic infiltration around bronchioles was seen at all time points.

***In vivo* efficacy studies on TR in guinea pig model of tuberculosis.** The dosing yielded around 98 viable bacilli per guinea pig lung as determined by plating the lungs of infected animals in their entirety one day post-infection. *M. tuberculosis* infection was allowed to develop for 15 days before the treatment started (Fig 4). No deaths were observed in either of the control groups of guinea pigs, or no significant change in mean body weight was seen in that group during the experiment. Animals were sacrificed at designated time points and lungs and tissue homogenate plated for CFU enumeration.

Dose-dependent loss in weight and appetite were observed among animals that were administered with TR. The conditions of animals deteriorated further after the third dose in 0.02 and 0.04 mg kg$^{-1}$ group and all animals were euthanized before schedule following humane end points guidelines after 3$^{rd}$ dose. Guinea pigs treated with 0.01 mg kg$^{-1}$ TR showed deuteriation in general body condition after the 5$^{th}$ dose; and one animal was euthanized on the next day after the 5$^{th}$ dose and 2 guinea pigs were euthanized after the 6$^{th}$ dose (S11 Table in S1 File). The experiment was terminated and remaining one animal was sacrificed post 48 hours of administering the 6$^{th}$ dose of TR.

**Effect of TR on TB burden in guinea pigs.** The bacterial load was enumerated by inoculating the tissue homogenates of each group onto the 7H11 medium supplemented with 10%

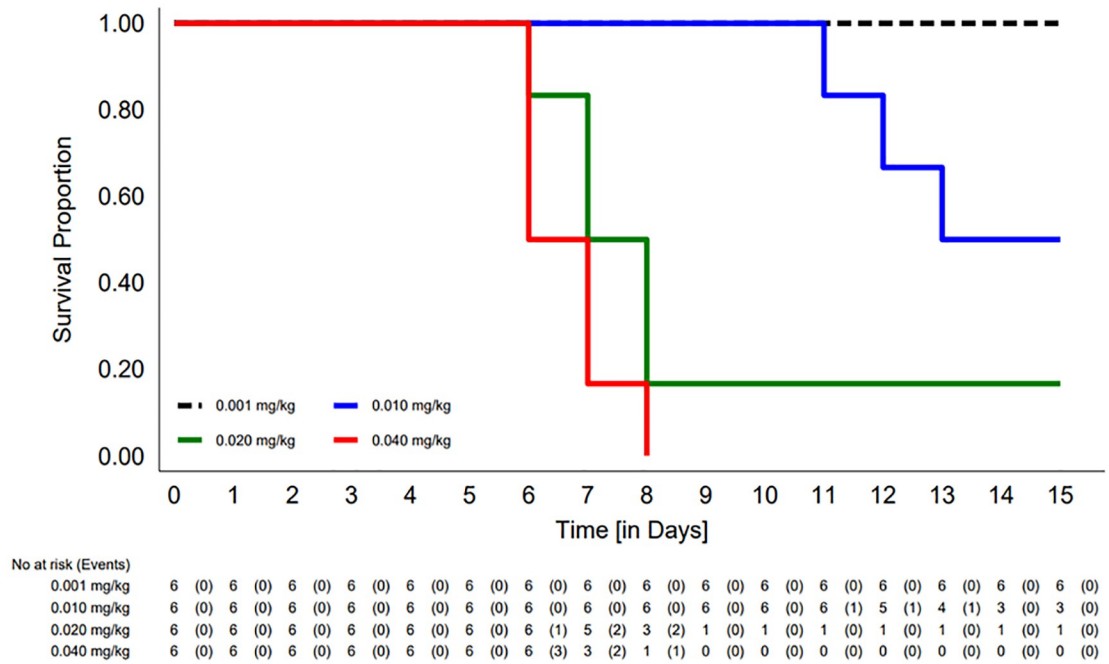

**Fig 4. Plot of Kaplan-Meier survival analysis (survival curve) for four groups of guinea pigs after Transitmycin administration is depicted.** In this study, the outcome event of mortality after third dose of drug was observed at concentrations with median survival time 0.01mg/kg 13[12-15days], 0.02mg/kg 7[7-8days] and 0.04mg/kg 6[6-7days] respectively. Further, all the animals survived at 0.001mg/kg (n = 6/6) concentration for 15 days.

OADC and antibiotics (Cyclohexamide 25µgml⁻¹, Amphotericin-B 10µg ml⁻¹ and Vancomycin 5µg ml⁻¹, Sigma). CFU was enumerated in each group that received TR, INH+RIF, erythromycin, and untreated control group. Post-aerosol infection bacterial load was found to be ~6.10 $\log_{10}$ CFU and ~6.4 $\log_{10}$ CFU at the time of study completion. Significant reduction in the CFU in the lung and spleen of guinea pigs treated with INH+Rif combination regimen. Guinea pigs that received 0.02 and 0.04 mg kg⁻¹ dose of TR did not show any reduction in CFU. Post-5th dose, the TR group that received 0.01mg kg⁻¹ dosage showed a reduction in bacterial load (4.9$\log_{10}$) only in the lung but not significantly lower when compared to the INH+RIF control group. However, infection load increased to 5.4 $\log_{10}$ with further dosage.

To reduce the toxicity observed at higher doses, the efficacy study at lower doses (0.001mg kg⁻¹) was performed and animals were sacrificed after 15th dose and 20th dose. There was no significant difference in the infection load between the treated and untreated group of animals that makes it clear that TR at 0.001mg kg⁻¹ body weight is not efficient in curtailing *M. tuberculosis* in the guinea pigs (S6A Fig in S1 File). On gross examination, necrotic lesions indicative of Mtb infection in the lung and splenomegaly suggestive of mycobacterial dissemination and inflammation were observed (S6B Fig in S1 File). Histopathological analysis of the lung revealed granuloma with no signs of drug toxicity.

**Pharmacokinetic (Pk) values of TR in uninfected guinea pigs.** Post TR drug administration, the level of TR in the serum of the uninfected guinea pigs was estimated. Initially, the Pk value was 0.5µgml⁻¹ and peaked at 4 hours of administration (0.21µg ml⁻¹) and as the time proceeds, it decreased to 0.09 µg ml⁻¹ after 24 hours (S7 Table in S1 File).

**Activity and properties (toxicity alerts) for TR by virtual screening.** Insilico study was carried out in four steps i) Similarity search by Cheminformatics tools ii) AI tools like SOM

**Table 1. Resemblance of Transitmycin with other closer analogues.**

| Compound | 2 dimension similarity (%) | 3 dimension similarity (%) |
|---|---|---|
| Actinomycin D | 86.66 | 64.04 |
| Actinomycin C2 | 89.53 | 69.90 |
| Actinomycin X2 | 100 | 74.4 |
| Rifampicin | 2.14 | 53.25 |

and clustering iii) Toxicity prediction by Tox alerts & algorithms iv) Structural understanding by molecular docking studies.

During the first phase, homology searches revealed that TR is a close analogue of several known molecules, as shown in the S8 Table in S1 File. In the second phase, similarity studies based on machine learning (ML), ANN (SOM), and clustering analysis revealed a close resemblance of the molecule with analogues that show similar toxic alerts [21], as shown in Table 1. Theoretical prediction using artificial neural networks and machine learning is a recognized cutting-edge method for medicinal chemistry and cheminformatics.

Such analogues include actinomycin X2, actinomycin-D etc. Toxic study by ML provided several alerts for DNA intercalators due to genotypic and mutagenic fragment presence in the TR and its analogues. To understand the toxic alert of TR due to intercalation with the DNA [22], docking analysis of TR and Actinomycin (with and without chromophore) were performed. Docking studies were performed by docking TR and actinomycin against the DNA of Mtb. Both TR and actinomycin exhibited similar glide score (-9.839) with chromophores while actinomycin showed more bondage with peptides. TR interactions with DNA were observed with the NH2 of chromophore and GC Base Pair of DNA. The type of interaction observed is Hydrogen Bonding and Pi-Pi bonding, but no peptide interactions were observed (S7 Fig in S1 File). Actinomycin exhibited a similar type of bonding with DNA (S9, S10 Tables in S1 File). Structure activity relationship (SAR) analysis was carried out further to confirm the DNA intercalation property. The analysis conducted shows that the TR is a DNA intercalator and the intercalation is caused due to the interaction of $NH_2$ and = O with the G-C base pair of the planar ring structure of the chromophore. Hence, it could be concluded that the anti-mycobacterial activity of TR is because of its DNA intercalation property.

Additionally, machine learning and AI-based analysis on the structural similarity of TR and its analogs with openly available anti-TB molecules were performed to check for any additional probable targets. The analysis performed was based on the physicochemical properties and pharmacophore descriptors of the molecules. The outcome of the analysis revealed two additional targets, namely recombinase A (recA) (PDB ID: 1MO3) and Methionine Aminopeptidases (PDB ID: 3IU8 and 3PKE) in addition to the DNA (S8 Fig in S1 File). High-throughput cheminformatics tools and bioinformatics methodologies are excellent tools for predicting experimental outcomes. In this study ML method was applied for similarity search. *In-silico* analysis revealed binding of the chromophore region alone, with the newly identified targets as in the case during DNA intercalation.

Because TR is toxic through in-vivo studies, in-silico prediction of structurally similar but less toxic TR analogs was attempted. A total of 57 such analogs were synthesized among which one analog (compound 47) (R = Hgrp) (S9 Fig in S1 File) showed less interaction with DNA (score = -4.618) and showed significant interaction with the 3IU8 (Score = -8.561).

*In-vitro* Ethidium Bromide (EtBr) displacement assay on TR and its analogs also showed intercalation with Mtb DNA in a dose-dependent manner (S10 Fig in S1 File) and confirmed

the *in-silico* analysis. As the TR and its analogs are competitively intercalated into the groves of DNA, it prevented EtBr from binding and thereby caused reduced fluorescence.

## Discussion

Unearthing any novel antibiotics brings new hope like an arrow in the quiver. TR at the time of its discovery from the culture filtrates of R2 appeared to be a light at the end of the tunnel of the TB therapy. Convincingly, TR exhibited significant anti-TB activity *in-vitro* and was found to be non-toxic in the *ex-vivo* studies. Hence carried further for pre-clinical trials to assess its safety and efficacy in the animal models. Through *in-silico* and Chemi-informatics approaches analogs of TR were designed and synthesized. Using column chromatography TR was purified up to 98.64%. MIC of TR was found to be between 5μg ml$^{-1}$ and 10μg ml$^{-1}$ against the maximum number of drug resistant clinical isolates of Mtb. The safety of TR was examined in three animal models, mice, rats, and guinea pigs, and the efficacy was tested in the guinea pig model.

The outcomes of the safety study and the TR was found to be highly toxic even at the dose of 0.005mg kg$^{-1}$ in guinea pigs. The lowest dose of TR at 0.001mg kg$^{-1}$ that was found safe and non-cytotoxic but did not protect the animals from disease progression with no significant variation in the infection load between TR treated and the untreated control group (S6A, S6B Fig in S1 File). Notably, TR at the dose of 0.01mg kg$^{-1}$ reduced the infection load initially, but with increased dose time points, the infection burden increased along with the events of worsened gross tissue pathology and hepato-renal damage (S11-S13 Figs in S1 File). The drift in the infection load pattern may be because of the toxic effects of TR that would have impacted the immune system of the guinea pig to take the pathogen side [23]. This observation is contradictory to the recommendation of a higher dose of anti-TB drugs to counterbalance the Mtbs anti-drug mechanism [24].

Analogs of TR; TR-SUND, TR-RB, TR-R, TR-NC6, FN-Me, and C-terbutyl showed limited anti-TB activity in-vitro compared to parental compound TR (Fig 5; S6 Table in S1 File). *In-silico* and Chemi-informatics studies were carried out to find out potential targets of the TR. Two targets of TR namely recA and methionine aminopeptidases were predicted based on the structural similarities with the existing anti-cancer drug actinomycin. Above and beyond TR was also found to intercalate into the groves of DNA. This predicted DNA intercalating property was proven to be true using an in-vitro EtBr displacement assay. This nature was exhibited by many antibiotics for instance DNA intercalating anthracyclin drugs were shown to be a potent anti-TB drug by binding with the Mtb primase DnaG gene, however, intercalation was shown to be directly associated with potency and inversely related to toxicity [25]. However DNA intercalators are dissuaded by World Health Organization [26] as TB treatment requires a lengthy regimen, and hence drugs with toxicity are to be avoided [27].

RecA, another target of TR is a recognized universal drug target in many pathogenic bacteria [28]. A compound named suramin [29] was found to target RecA protein and to repress the expression of the recA gene as well. RecA controls the expression of genes that promotes survival post-DNA damage. Besides, RecA is involved in various other physiological processes of Mtb that include the synthesis of toxins and other virulence factors [29]. However, though RecA is considered as a potential drug target, any novel drug with RecA targeting ability should be devoid of interaction with the RecA homolog Rad51. Methionine aminopeptidases remove N-terminal methionine through its metalloprotease property during protein synthesis and any loss in its function results in the reduced Mtb viability [30]. Previously parental Bengamide compound was found to non-specifically target methionine aminopeptidases of both humans and Mtb hence attempts were made to synthesize its derivatives with less cytotoxicity [31].

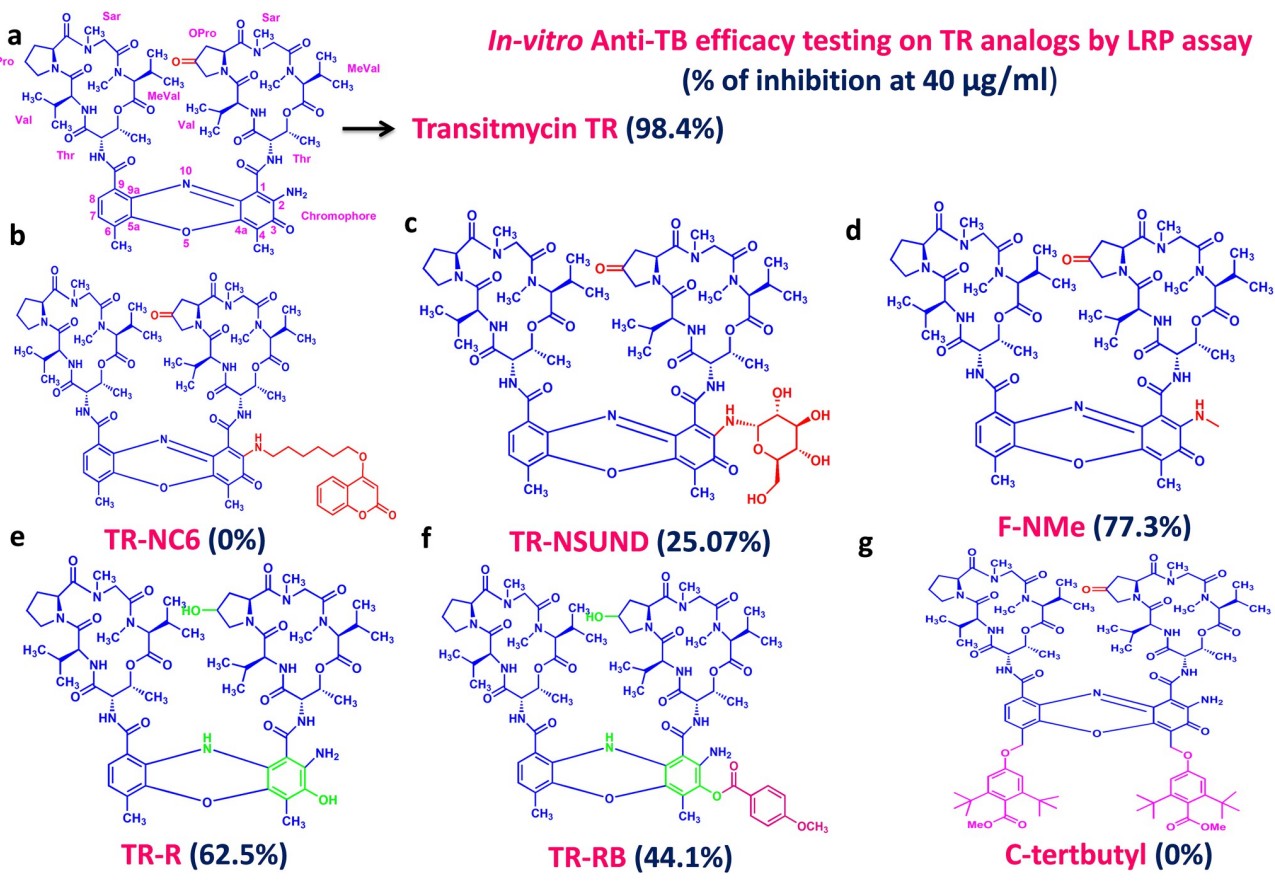

**Fig 5. In-vitro anti-TB efficacy testing on TR analogues by LRP assay.**

TR hits the above three targets however it is not specific against Mtb especially in the case of DNA intercalation that was proved through this study both by *in-vitro* and *in-silico* methods. Hence, at this stage, TR and its newly synthesized analogs could not be considered for human therapeutic use. However, as TR is structurally similar to actinomycin that is being used as an anti-cancer drug, TR can either be evaluated further for anti-cancer property or the new structural analogs that are free of DNA intercalation property can be synthesized with the aid of advanced Chemi-informatics tools. Latterly, using SARs analysis, a new compound number 47 (S9 Fig in S1 File) is proposed to have lesser toxicity and high functional potency. Our studies have shown TR intercalation into DNA is mediated via bonding through $-NH_2$ and $= O$ of chromophore's planar ring structure with GC base pair. Consequently, this opens up the scope for lead optimization of TR to gain a newer analog that holds Chemical absorption, distribution, metabolism, excretion, and toxicity (ADMET) properties.

## Supporting information

**S1 File. Contains all the supporting tables.**
(DOCX)

## Author Contributions

**Conceptualization:** Rajesh Mondal, Azger Dusthackeer V. N., Dinesh Bharadwaj, Suresh Ganesan, Hemanth Kumar A. K., Mukesh Doble, Vanaja Kumar.

**Data curation:** Azger Dusthackeer V. N., Kannan Thiruvengadam, Suresh Ganesan, Mukesh Doble, Vanaja Kumar.

**Formal analysis:** Palaniyandi Kannan, Mahizhaveni Balasubramanian, Sam Ebenezer Rajadas, Mukesh Doble, Balagurunathan R., Vanaja Kumar.

**Funding acquisition:** Azger Dusthackeer V. N., Radhakrishnan Manikkam, Manjula Singh, Jaleel U. C. A., Balagurunathan R., Srikanth Prasad Tripathy, Vanaja Kumar.

**Investigation:** Amit Kumar Singh, Jaleel U. C. A., Mukesh Doble, Vanaja Kumar.

**Methodology:** Palaniyandi Kannan, Radhakrishnan Manikkam, Shainaba A. S., Mahizhaveni Balasubramanian, Padmasini Elango, Jaleel U. C. A., Balagurunathan R.

**Project administration:** Azger Dusthackeer V. N., Dinesh Bharadwaj, Hemanth Kumar A. K., Shripad Patil, Jaleel U. C. A., Mukesh Doble, Balagurunathan R., Vanaja Kumar.

**Resources:** Jaleel U. C. A., Balagurunathan R., Vanaja Kumar.

**Software:** Hemanth Kumar A. K.

**Supervision:** Rajesh Mondal, Dinesh Bharadwaj, Jaleel U. C. A., Mukesh Doble, Srikanth Prasad Tripathy, Vanaja Kumar.

**Validation:** Palaniyandi Kannan, Kannan Thiruvengadam, Padmasini Elango, Dinesh Bharadwaj, Gandarvakottai Senthilkumar Arumugam, Hemanth Kumar A. K., Shripad Patil, Mukesh Doble, Balagurunathan R., Vanaja Kumar.

**Visualization:** Azger Dusthackeer V. N., Padmasini Elango.

**Writing – original draft:** Azger Dusthackeer V. N., Padmasini Elango, Dinesh Bharadwaj, Mukesh Doble, Vanaja Kumar.

**Writing – review & editing:** Azger Dusthackeer V. N., Sam Ebenezer Rajadas, Dinesh Bharadwaj.

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
