## [Decision Letter · Decision Letter 0]

2 Sep 2022

PONE-D-22-18915In-vivo studies on Transitmycin, a potent Mycobacterium tuberculosis inhibitorPLOS ONE

Dear Dr. Dusthackeer,

Thank you for submitting your manuscript to PLOS ONE. After careful consideration, we feel that it has merit but does not fully meet PLOS ONE’s publication criteria as it currently stands. Therefore, we invite you to submit a revised version of the manuscript that addresses the points raised during the review process.

Please submit your revised manuscript by October 17, 2022. If you will need significantly more time to complete your revisions, please reply to this message or contact the journal office at plosone@plos.org. Please include the following items when submitting your revised manuscript:A rebuttal letter that responds to each point raised by the academic editor and reviewer(s). You should upload this letter as a separate file labeled 'Response to Reviewers'.A marked-up copy of your manuscript that highlights changes made to the original version. You should upload this as a separate file labeled 'Revised Manuscript with Track Changes'.An unmarked version of your revised paper without tracked changes. You should upload this as a separate file labeled 'Manuscript'.

We look forward to receiving your revised manuscript.

Kind regards,

Frederick Quinn

Academic Editor

PLOS ONE

Journal Requirements:

"Authors thanks the Indian Council of Medical Research for providing 

 grant-in-aid (F.No.5/8/5/8/TF/2017/ECD-I) to conduct this preclinical study on Transitmycin".

"The funders had no role in study design, data collection and analysis, decision to publish, or preparation of the manuscript".

6. Please include a copy of Tables 1 and 6 which you refer to in your text on pages 7 and 19, respectively.

Reviewers' comments:

Reviewer's Responses to Questions

**Comments to the Author**

1. Is the manuscript technically sound, and do the data support the conclusions?

Reviewer #1: Partly

Reviewer #2: Yes

2. Has the statistical analysis been performed appropriately and rigorously? 

Reviewer #1: N/A

Reviewer #2: No

3. Have the authors made all data underlying the findings in their manuscript fully available?

Reviewer #1: No

Reviewer #2: No

4. Is the manuscript presented in an intelligible fashion and written in standard English?

Reviewer #1: No

Reviewer #2: No

5. Review Comments to the Author

Reviewer #1: In general, this manuscript is badly prepared, the authors need to take the submission of their work seriously. There are lots of issues the authors need to seriously address, below are some but not all:

1. Line 231, please keep the citation format consistent.

2. Line 272, what is exact range of <0.312 and 40, 0 is also <0.312, this way of expression is very unprofessional.

3. Line 273, A maximum of 19 and 15 isolates got inhibited at a 5μg ml-1and 10μg ml-1concentration of TR respectively. Why a lower concentration has a higher number of inhibited isolated?

4. The authors did not properly assign and arrange figures and supplementary figures, I see a number of supplementary figures while only a few figures in main text.

5. The authors did a terrible job in preparing figures. In figure1, what are the numbers under the X axis? What is the Y axis? What are the units? two drug susceptible isolates of M. tuberculosis, which one is the first one, which is the second? I have a difficulty in understanding this figure 1, the authors definitely need to explain more detail in the figure legend.

6. From the figure 1, if my interpretation is correct, this TR chemical is not even as efficient as kanamycin, what's the significance of the usage of this chemical to treat TB?

7. Figure 2 has no units and detailed figure legend.

8. Figure 3, at least the authors should keep the style, format, such as font and size consistent. In addition to this issue, no legend. What are G1 G2 and G3, why G2 and G3 stopped after week5 or 6?

9. Line 332, the authors need to write the numbers in a more scientific format, what is the number of ~6.10 log10? The authors need to go through the entire text and figures and tables to list numbers in a scientific way.

10. Line 351, what is PK?

11. Line 357, a docking analysis is not able to confirm anything, it is a simple prediction.

12. Line 369. Hence it could be concluded that the anti-mycobacterial activity of TR is because of its DNA intercalation property. I do not see sufficient data to conclude this, docking prediction analysis does not count.

13. Line 380, the analogs interaction with DNA needs to be experimentally tested but not just by prediction.

14. Line 421, how do the authors know RecA is a target of TR, again the prediction analysis does not serve as an evidence.

15. Missing reference for suramin.

16. Line 433, the authors do not have experimental data suggesting TR hits the above 3 targets.

17. Line 590, what is supplementary figure1, please do not simply crop the image from somewhere else without even erasing extra contents. And again, the authors should learn how to write a proper figure legend to explain figure contents.

18. Supplementary figure 4, 5 and 10 are redundant to main figures.

19. Supplementary figure 6, what are a b,c? Simply stacking figures while not explaining them will not serve well to explain your study.

20. Supplementary figure 14, what are the organs?

Reviewer #2: 1. Please provide the source of all the chemicals and biologicals (bacteria, animals)

2. L98 I am not sure if the ratio is written correctly ( 100%:1%)

3. there are two sets of supplementary figures and tables with this manuscript. One at the end of it and another as a supplementary file. This created a lot of confusion for me. Please combine them into one and number them correctly. A lot of supp figures and table were not referenced in the manuscript.

4. Please provide a figure with TR and all its analogues in the manuscript

5. L127. Provide reference for broth microdilution method here

6. Please provide a brief description of the LRP assay

7. Need more description about the Pharmacokinetic study on TR method

8. L164, Possible typo, the peritoneal route?

9. L169, typo, titered.

10. L198, should be supp fig 2.

11. L205, please refer to supp fig 3. Fig needs to be reformatted.

12. L246, which software was used? How the analysis was done. Need more information for the methods.

13. L249, how the fluorescence was collected? Plate reader/ fluorimeter etc?

14. Fig 1 and 2 needs more information in the legend. In its current form I can not understand what any of these figure means. What is in X and Y axes? Probably need more detail method about the experiment.

15. Paragraph 287, Please summarize the result in a table or figure

16. Figure 3 needs more information in the legend. What are those lines? What is G1, G2 G3?

17. L306, 308. Again wrong figure was referenced here

18. L308, I think there is a figure missing here for histopathological examination.

19. L313 wrong fig referenced. Some the bar graphs have no error bars

20. L338 wrong figure cited

21. L347, 348 only headings?

22. L355, what kind of basic analysis showed this?

23. L363 wrong figure

6. PLOS authors have the option to publish the peer review history of their article (what does this mean?). If published, this will include your full peer review and any attached files.

Reviewer #1: No

Reviewer #2: No

---

## [Author Response · Author response to Decision Letter 0]

18 Oct 2022

We thank all of you for your time and effort in reviewing our manuscript. As researchers, we have worked tirelessly to highlight Transitmycin as a potent anti-TB drug.  This paper will undoubtedly assist the current and future research community in conducting experiments on newer drug compounds. We personally thank you for providing us with another opportunity to revise our manuscript based on your valuable suggestions. In the revised manuscript, we explained all of the comments. We humbly request that you encourage us to proceed with publication.

Editor comments: 

1. Your ethics statement should only appear in the Methods section of your manuscript. If your ethics statement is written in any section besides the Methods, please delete it from any other section. 

Ans. Ethics statement is changed from previous section to Methods section.

2. Please include a copy of Tables 1 and 6 which you refer to in your text on pages 7 and 19, respectively

Ans. The table are not included in the current version of manuscript and removed from the main text.

3. Funding disclosure: 

Ans. Funding disclosure has been removed from the manuscript and funding details were given in the Funding statement form in the online submission.

---

## [Decision Letter · Decision Letter 1]

6 Jan 2023

PONE-D-22-18915R1In-vivo studies on Transitmycin, a potent Mycobacterium tuberculosis inhibitorPLOS ONE

Dear Dr. Dusthackeer,

Thank you for submitting your manuscript to PLOS ONE. After careful consideration, we feel that it has merit but does not fully meet PLOS ONE’s publication criteria as it currently stands. Therefore, we invite you to submit a revised version of the manuscript that addresses the points raised during the review process.

In accordance with PLOS ONE’s submission guidelines pertaining to humane endpoints (http://journals.plos.org/plosone/s/submission-guidelines#loc-humane-endpoints), please provide further details the humane end points experimental procedures. In particular please provide more clarity on the 6 experimental groups used, a survival curve for this study, the rationale for performing the infection/efficacy study reported in Figure S2, and specify if any animals died before meeting the criteria for euthanasia. 

We look forward to receiving your revised manuscript.

Kind regards,

Fred Quinn

Academic Editor

PLOS ONE

Additional Editor Comments:

In accordance with PLOS ONE’s submission guidelines pertaining to humane endpoints (http://journals.plos.org/plosone/s/submission-guidelines#loc-humane-endpoints), please provide further details the humane end points experimental procedures. In particular please provide more clarity on the 6 experimental groups used, a survival curve for this study, the rationale for performing the infection/efficacy study reported in Figure S2, and specify if any animals died before meeting the criteria for euthanasia. 

Reviewers' comments:

Reviewer's Responses to Questions

**Comments to the Author**

1. If the authors have adequately addressed your comments raised in a previous round of review and you feel that this manuscript is now acceptable for publication, you may indicate that here to bypass the “Comments to the Author” section, enter your conflict of interest statement in the “Confidential to Editor” section, and submit your "Accept" recommendation.

Reviewer #2: (No Response)

2. Is the manuscript technically sound, and do the data support the conclusions?

Reviewer #2: Yes

3. Has the statistical analysis been performed appropriately and rigorously? 

Reviewer #2: Yes

4. Have the authors made all data underlying the findings in their manuscript fully available?

Reviewer #2: Yes

5. Is the manuscript presented in an intelligible fashion and written in standard English?

Reviewer #2: Yes

6. Review Comments to the Author

Reviewer #2: The authors have responded to all my concerns. However a rebuttal letter with mentioning all the changes would be beneficial.

7. PLOS authors have the option to publish the peer review history of their article (what does this mean?). If published, this will include your full peer review and any attached files.

Reviewer #2: No

---

## [Author Response · Author response to Decision Letter 1]

26 Jan 2023

We have included humane end points followed during animal study and also inserted a figure (Fig 4) depicting the survival curve of the guinea pigs used. We have clearly mentioned on the experimental groups of animals, the rationale for performing the infection/efficacy study reported in Figure S2 has been addressed along with the criteria for euthanasia. We hope we have given sufficient information on data required by the reviewers and accept our manuscript for publication.

---

## [Decision Letter · Decision Letter 2]

16 Feb 2023

In-vivo studies on Transitmycin, a potent Mycobacterium tuberculosis inhibitor

PONE-D-22-18915R2

Dear Dr. Dusthackeer,

We’re pleased to inform you that your manuscript has been judged scientifically suitable for publication and will be formally accepted for publication once it meets all outstanding technical requirements.

Kind regards,

Frederick Quinn

Academic Editor

PLOS ONE

Additional Editor Comments (optional):

Reviewers' comments:

Reviewer's Responses to Questions

**Comments to the Author**

1. If the authors have adequately addressed your comments raised in a previous round of review and you feel that this manuscript is now acceptable for publication, you may indicate that here to bypass the “Comments to the Author” section, enter your conflict of interest statement in the “Confidential to Editor” section, and submit your "Accept" recommendation.

Reviewer #2: All comments have been addressed

2. Is the manuscript technically sound, and do the data support the conclusions?

Reviewer #2: Yes

3. Has the statistical analysis been performed appropriately and rigorously? 

Reviewer #2: Yes

4. Have the authors made all data underlying the findings in their manuscript fully available?

Reviewer #2: Yes

5. Is the manuscript presented in an intelligible fashion and written in standard English?

Reviewer #2: Yes

6. Review Comments to the Author

Reviewer #2: All my questions and comments were previously addressed and I don't have any further questions regarding this manuscript.

7. PLOS authors have the option to publish the peer review history of their article (what does this mean?). If published, this will include your full peer review and any attached files.

Reviewer #2: No

---

## [Editor Report · Acceptance letter]

21 Feb 2023

PONE-D-22-18915R2 

In-vivo studies on Transitmycin, a potent *Mycobacterium tuberculosis* inhibitor 

Dear Dr. Dusthackeer VN:

I'm pleased to inform you that your manuscript has been deemed suitable for publication in PLOS ONE. Congratulations! Your manuscript is now with our production department. 

Kind regards, 

on behalf of

Dr. Frederick Quinn 

Academic Editor

PLOS ONE